# A Statistical Method for Attack-Agnostic Adversarial Attack Detection with Compressive Sensing Comparison

## Abstract

Adversarial attacks present a significant threat to modern machine learning systems. Yet, existing detection methods often lack the ability to detect unseen attacks or detect different attack types with a high level of accuracy. In this work, we propose a statistical approach that establishes a detection baseline before a neural network's deployment, enabling effective real-time adversarial detection. We generate a metric of adversarial presence by comparing the behavior of a compressed/uncompressed neural network pair. Our method has been tested against state-of-the-art techniques, and it achieves near-perfect detection across a wide range of attack types. Moreover, it significantly reduces false positives, making it both reliable and practical for real-world applications.

## 1 Introduction

Neural networks (NNs) are being used in numerous use cases across many disciplines. They have almost become an integral part of our lives. One variation of them that we use very commonly is the Convolutional Neural Network (CNN). They are widely used in image recognition, and they are considerably reliable in this application, and they keep getting better virtually every day. A challenge in using CNNs in sensitive applications is the existence of adversarial attacks (Goodfellow et al., 2015).

Adversarial attacks inject a small, ideally human-imperceptible perturbation in the input image before being fed into a CNN classifier. This perturbation usually takes the form of random noise and is applied at a practically imperceptible level. Although this modification looks harmless or even completely invisible to the human eye, it wreaks havoc inside the workings of the CNN classifier. It ultimately pushes the detection to an invalid class. This can lead to many unseemly outcomes, ranging from loss of accuracy to failure of safety-critical systems.

Several methods exist to detect and suppress adversarial attacks. However, these methods suffer from inherent flaws, such as the requirement of apriori knowledge of the attack type, the high number of false positives, low overall accuracy, and scaling issues with different network architectures.

In this paper, we present a simple attack-agnostic detection method that does not require prior knowledge of attack models. It requires a simple training process before the deployment to generate a set of class identities and, during runtime, uses those identities to match every incoming sample.

Compression suppresses adversarial noise to an extent, as shown in Aydemir et al. (2018) among others. While this effect is less than ideal for reliably suppressing all adversarial perturbations, we can observe a difference when we take the same input and run it through a pair of slightly different networks. The pair of networks will be almost identical, except that to the second network, we feed a compressed version of the image, and the network itself is pre-trained on compressed images after regular training. We leverage a secondary denoising network that operates on compressed images and check how far the matching of the distributions skew between the two networks. We propose a metric that can be calculated at runtime for each sample that measures this disparity and a threshold $T$ that can be empirically determined pre-deployment. We use the threshold on the metric to determine the presence of adversarial perturbations and filter out the adversarial samples.

The metric calculation uses the feature maps generated in both networks' last feature layer (the layer before the winner-takes-all/softmax layer) and represents how much they disagree. This disagreement is more pronounced in adversarial images, thus allowing the discrimination between them and clean images. Our method consistently gives accurate detections for every adversarial attack tested, while existing work performs well in some attacks and poorly in others.

We developed this method without considering any of the attack models and their behavior since compressive networks suppress almost any adversarial signal presence. This led us to believe this method should perform well on any attack universally. We claim that our method is an attack-agnostic adversarial attack detector.

Several mathematical/statistical operations are used in this work. One of the notable operators is the KL divergence, which is defined for two vectors $\mathbf{a} = (a_1, a_2, ..., a_n)$ and $\mathbf{b} = (b_1, b_2, ..., b_n)$ as follows, where $ln$ stands for the natural logarithmic operator.

$$KL(\mathbf{a}, \mathbf{b}) = \sum_{i=1}^{n} a_i . ln(\frac{a_i}{b_i}) \tag{1}$$

One important point to note here is that the KL operator is not commutative, and therefore, $KL(a, b)$ and $KL(b, a)$ are not necessarily equal.

We utilize the L2 norm to measure the difference between two vectors. The L2 norm between two vectors $a$ and $b$ are defined as,

$$L2\_norm(a, b) = \sqrt{\sum_{i=1}^{n} (a_i - b_i)^2} \tag{2}$$

Also, we used the Mann-Whitney U test to compare two smaller distributions and decide whether the data belongs to a common distribution. This method is outlined in Mann & Whitney (1947)

In the rest of the paper, section 2 briefly outlines related work. Section 3 presents the details of the technique and how it works. Our experimental results are showcased in section 4, and we compare our approach with existing methods. Finally, section 5 concludes the paper.

## 2 RELATED WORK

We use several adversarial attack models to verify our theory of attack agnostic detection. They are contained within the Adversarial Robustness Toolbox (ART) by Nicolae et al. (2018) Python package. The attack models that are used here are the Fast Gradient Sign Method (FGSM) (Goodfellow et al., 2015), the Projected Gradient Descent (PGD) (Madry et al., 2017) approach, the Square Attack (Andriushchenko et al., 2020) method, the DeepFool (Moosavi-Dezfooli et al., 2015) method and the Carlini-Wagner(CW) (Carlini & Wagner, 2016) attack. These methods use an approximately similar CNN model (black box attack) or the exact CNN model used in detection (white box attack) to generate an attack. The exact way they create the attack varies by the attack method. Still, they usually use gradient-based methods, where the adversary calculates the gradient of the model's loss function with respect to the input and adjusts the input accordingly to maximize the loss, as opposed to minimizing the loss when the goal is to predict the image content accurately. A noise vector is calculated using these gradients that skew the prediction in a way that increases the chance of mis-classification. This noise vector is then added to the image, making it fool the detector. The idea of our work is to identify images that have been tampered with with such a malicious noise vector.

Current methods of detecting the presence of adversarial attacks include attack agnostic methods such as the Least Significant Component Feature (LCSF) (Cheng et al., 2022) method, the Energy Distance/Maximum mean discrepancy (Saha et al., 2019) method, the Mahalanobis distance-based classifier (Lee et al., 2018) method. Also, there exist attack specific methods such as the Feature Squeezing (Xu et al., 2018) method, using Latent Neighborhood Graphs (Abusnaina et al., 2021), using Influence Functions/Nearest Neighbors (Cohen et al., 2020), using Bayesian Neural Networks (Deng et al., 2021), by random input responses (Huang et al., 2019) and by Random Subspace

Analysis (Drenkow et al., 2021). These methods have their shortcomings, such as poor performance in some attack models, requiring extra training data, high false positive rates, and limited flexibility.

One major part of our method is the secondary denoising network. In theory, this can be accomplished in numerous methods, but we have chosen compressive sensing using JPEG2000 compression, as demonstrated by Aydemir et al. (2018). JPEG compression is typically used to reduce the file size of images by reducing unnecessary and imperceptible information contained in an image. However, this provides a benefit when attempting to suppress adversarial attacks since the JPEG compression algorithm treats the adversarial noise signals as imperceptible information and disregards them, restoring the original image to an extent. The downside of this in practice is that the accuracy improvement is not perfect and, in most cases, only restores about 20-30% of the accuracy. So, this alone is not a comprehensive defense strategy against adversarial attacks. However, since we know that a compressed network treats adversarial noise differently, we can take advantage of that to implement a more sophisticated detection method.

In order to properly build class identities and match new samples, we need a proper comparison system. This is accomplished using the method proposed by Pentsos & Tragoudas (2023). The idea is to partition the pre-provided train-test sets and use the feature vectors of those partitions to build an identity. Then, the new samples are augmented using benign noise vectors to form a rich representation of the sample, which is compared against the pre-built class identities. We use this underlying concept to check how the new samples match our known class baselines.

## 3 METHODOLOGY

Here, we describe a method to detect adversarial attacks in an attack-agnostic way, using a pre-built class identity and the deviation from it on a new sample. The sample is run through the regular network and a redundant network, which uses a denoising method such as JPEG compression Aydemir et al. (2018). Before the deployment, we run the system through a known dataset and build each class's identities on both networks. Then, in the field, we match each example to the class identity on both networks and extract a measure of how much the two networks disagree on the image. If they disagree beyond a certain threshold, the sample is marked adversarial. This effect is evident in the sample images shown in Figure 1

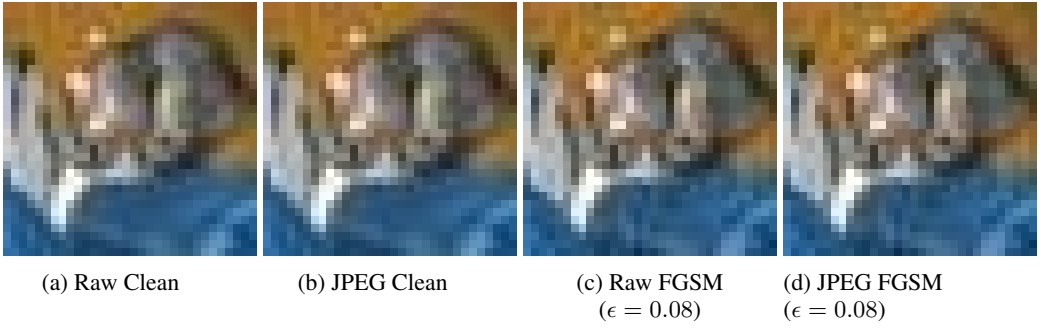

   (a) Raw Clean         (b) JPEG Clean        (c) Raw FGSM       (d) JPEG FGSM
                                                ($\epsilon = 0.08$)        ($\epsilon = 0.08$)

Figure 1: Sample image of the class 'Cat' from CIFAR-10 at the various stages of the method

Our method is derived from the technique from Pentsos & Tragoudas (2023) to build and match class identities for a classification problem. This allows us to have an apriori knowledge of the data, how each class behaves, and a way to identify when they misbehave, i.e., an adversarial attack. We used the Mann-Whittney U test and the KL divergence to generate and match distributions with each other. For the identity creation phase, we run Algorithm 1 to create the class identity for each class. This takes the form of a histogram, one for each class. The second phase generates a distribution from the input image using Algorithm 2, matching them with each class distribution, getting the distance to the closest distribution, comparing how the redundant network agrees with this measure, and making the decision.

We preserve the name convention used in Pentsos & Tragoudas (2023) for better clarity. The algorithm 1 is used to calculate the distribution identity of each class and store it for later use. It takes

**Input** : Class $c$, test set of class $c$, training set of $c$, $I$ parameter
**Output:** Distribution identity $MW\_id(c)$ of class $c$
**for** $i \leftarrow 1$ **to** $I$ **do**
    Randomly sample $N$ images from the test set of $c$ to create sample $S_T$;
    Partition $S_T$ into subsets $S_i$, each containing $k$ images;
    Randomly sample $N$ images from the train set of $c$ to create sample $S_R$;
    Perform a forward pass of $S_R$ through the CNN;
    **for** *each subset $S_i$ of $S_T$* **do**
        Perform a forward pass of $S_i$ through the CNN;
        Extract the average feature vectors $\hat{f}_n^{S_i}$ and $\hat{f}n^{S_R}$ of the examined samples using the output of the penultimate layer $n$ of the CNN;
        Normalize the two extracted average feature vectors;
        Calculate the KL divergence $D_{TR}^{(I)}$ between $\hat{f}_n^{S_i}$ and $\hat{f}n^{S_R}$
    **end**
    Partition $S_R$ into subsets $S_i$, each containing $k$ images;
    Perform a forward pass of $S_T$ through the CNN;
    **for** *each subset $S_i$ of $S_R$* **do**
        Perform a forward pass of $S_i$ through the CNN;
        Extract the average feature vectors $\hat{f}_n^{S_i}$ and $\hat{f}n^{S_T}$ of the examined samples using the output of the penultimate layer $n$ of the CNN;
        Normalize the two extracted average feature vectors;
        Calculate the KL divergence $D_{RT}^{(I)}$ between $\hat{f}_n^{S_i}$ and $\hat{f}n^{S_T}$
    **end**
    Calculate the p-value of the Mann-Whitney U test between $D_{TR}^{(i)}$ and $D_{RT}^{(i)}$;
    Store the p-metric as the $i$-th element of the distribution identity of class $c$;
**end**

**Algorithm 1:** Calculating the distribution identity $MW\_id$ of a class using the KL divergence

the train set and test set for each class as input, along with the parameter $I$, and outputs a dictionary of class maps for each class. The parameter $I$ is empirically determined by experimentation.

## 3.1 BUILDING THE DISTRIBUTION IDENTITY

When building the distribution identity for a class, we first take the test set $S_T$ for that class and partition it into subsets of size $k$ each, which we will call $S_i$. Then, we select $N$ random images from the train set to create the sample batch $S_R$. These image sets $S_R$ and each $S_i$ are passed through the neural network in question, as well as the feature vector of the layer before the softmax layer is extracted. We calculate the average vector of each of these sets, which we call $\hat{f}^{S_R}$ and $\hat{f}^{S_i}$ respectively. The average feature vector here is the element-wise average of each vector from the neural network output. These vectors are then normalized, and their KL divergence $D_{S_i S_R}$ is calculated as shown below.

$$D_{S_i S_R} = \frac{KL(\hat{f}^{S_i}, \hat{f}^{S_R}) + KL(\hat{f}^{S_R}, \hat{f}^{S_i})}{2} \tag{3}$$

We repeat this process for each subset $S_i$ of $S_T$, and then we are left with a series of 'divergence points', namely $D_{TR}$. We need to calculate $D_{RT}$, which involves the same procedure but with $S_R$ and $S_T$ interchanged. We then take these two populations and perform a Mann-Whitney U test (Mann & Whitney, 1947) between them. The resulting p-value is stored as the $i$th value of the class distribution identity. This process is repeated over $I$ iterations to build the complete distribution identity. The samples generated during the algorithm's execution are saved as a dictionary, *class_distributions*. This step is executed before the actual deployment of the neural network, and we need to make sure that the train and test sets are strictly free of perturbations, adversarial or random. This process is also performed separately on the redundant network.

## 3.2 DETECTING ADVERSARIAL SAMPLES

After the neural network is deployed, we use the second procedure, as outlined in algorithm 2, to determine whether the given image is adversarial. The algorithm takes in the unknown image $q$, the sample dictionary $class\_distributions$, and the class identity dictionary $MW\_id$. It will return the distance metric, which we can use to compare with a predetermined threshold, where if it is higher, the image will be tagged as adversarial and as clean otherwise. We take the input image and generate a sequence of $N$ images. To do this, we add various random noise signals to the image, save it as a new image, and generate $N-1$ images, which gives us $N$ images when combined with the original image. We call this batch of $N$ images as $S_q$, and it is a rich representation of the image itself.

Then, to match this new representation with the classes, we pick $N$ images from the class distribution for a given class $c$. These images will form our $S_T$ for this execution. We run this $S_T$ and $S_q$ through the same partitioning, p-value calculation, and KL divergence calculation we performed in algorithm 1. The resulting KL divergence value for each class is stored as a vector with respect to the class. Two such vectors are extracted, one from the standard network and the other from the redundant network.

---

**Input** : Input image $q$, dictionary $class\_distributions$, list of class distribution identities $MW\_id$
**Output:** Adversarial Possibility Metric $P_A$
Run $Instance(q, N)$, which creates $N-1$ uncorrelated instances of $q$ and form sample $S_q$
  with these $N$ images;
Partition $S_q$ into subsets of size $k$;
**for** *each class label c do* **do**
    Randomly select m samples from the distribution $class\_distributions[c]$;
    Forward pass images in $m$ through the deployed network and extract their average feature
      vector, denoted by $\hat{f}_c^m$;
    **for** *each subset S of $S_q$ do* **do**
        Forward pass images in S through the deployed network and extract their average
          feature vector, denoted by $\hat{f}_n^{sq}$;
        Append $p\_value$ between $\hat{f}_c^m$ and $\hat{f}_n^{sq}$ to the class-sample signature $D_c$
    **end**
    Compute the KL divergence between $D_c$ and $MW\_id(c)$ and append it to distance vector
    $V$;
**end**
Get $V$ for the raw network $V_R$ and for the compressed network $V_C$;
**Return** $L2\_Norm(V_R, V_C)$ as $P_A$;

**Algorithm 2:** Calculating the adversarial possibility metric

---

The two networks are used here because the denoising network will try to push back against the adversarial features and diminish them while pushing the image toward its original class and its resultant features. This will cause a disparity between the distribution distances between the class identities. The distance vector will match very closely on clean images but will take a significant value on the adversarial, giving a clean separation between the two.

We decide the detection threshold $T$ empirically. It is chosen so that all the clean samples score below this threshold and still give good results on the adversarial data. Once this universal threshold is established, it does not need to vary by the type of attack. Anything below is classified as clean, and anything above is classified as an adversarial sample. In this way, we can achieve true attack-agnostic detection.

## 4 EXPERIMENTAL RESULTS

We performed experiments using the proposed method on three datasets across five attacks. The chosen datasets are CIFAR-10, CIFAR-100, and a truncated version of Imagenet, which contains only 50 classes. FGSM, PGD, Square Attack, DeepFool, and Carlini-Wagner attacks were performed on them. Most of the evaluation was conducted on a workstation running Ubuntu 22.04 LTS, an Intel

Table 1: Effect of Compression on Adversarial Images

| Dataset | Raw Clean | Compressed Clean | Raw FGSM | Compressed FGSM |
|---------|-----------|------------------|----------|-----------------|
| CIFAR-10 | 85.63 | 83.65 | 62.72 | 81.59 |
| CIFAR-100 | 82.34 | 79.86 | 53.92 | 71.82 |
| ImageNet | 78.98 | 73.47 | 51.65 | 67.44 |

Table 2: Performance Comparison of Various Defenses

| Attack | Dataset | Dist. Matching[1] (ours) | Cheng[1] | Saha[1] | Mahalanobis | Feat. Squeezing | Abus-naina | Cohen | LiBre | Huang |
|--------|---------|--------------------------|----------|---------|-------------|-----------------|------------|-------|-------|-------|
| FGSM | CIFAR-10 | **100.00**[2] | 99.90 | 75.90 | 99.94 | 20.80 | 99.88 | 87.75 | -[3] | 77.20 |
| | CIFAR-100 | **100.00** | 100.00 | - | 99.86 | - | - | 87.23 | - | - |
| | ImageNet | **100.00** | - | - | - | 99.60 | 99.53 | - | **100.00** | - |
| PGD | CIFAR-10 | **100.00** | **100.00** | - | - | - | 91.39 | 99.34 | - | 96.40 |
| | CIFAR-100 | **100.00** | 99.90 | - | - | - | - | 81.87 | - | - |
| | ImageNet | *98.80*[4] | - | - | - | - | 99.35 | - | **99.40** | - |
| Square Attack | CIFAR-10 | *98.00* | - | - | - | - | **98.82** | - | - | - |
| | CIFAR-100 | **100.00** | - | - | - | - | - | - | - | - |
| | ImageNet | **99.50** | - | - | - | - | 82.20 | - | - | - |
| DeepFool | CIFAR-10 | **100.00** | 84.60 | - | 83.41 | 77.40 | - | 97.98 | - | 99.80 |
| | CIFAR-100 | **97.50** | 73.30 | - | 77.57 | - | - | 78.82 | - | - |
| | ImageNet | **99.50** | - | - | - | 78.60 | - | - | - | - |
| Elastic Net | CIFAR-10 | **98.90** | - | - | - | - | - | 86.95 | - | 95.10 |
| | CIFAR-100 | **97.30** | - | - | - | - | - | 70.49 | - | - |
| | ImageNet | **97.80** | - | - | - | - | - | - | - | - |
| CW | CIFAR-10 | 97.80 | 94.30 | **100.00** | 87.31 | 98.10 | 91.51 | 98.98 | - | 98.70 |
| | CIFAR-100 | **99.50** | 81.60 | - | 91.77 | - | - | 93.16 | - | - |
| | ImageNet | *97.10* | - | - | - | 97.90 | 86.05 | - | **98.50** | - |
| JSMA | CIFAR-10 | *98.60* | - | - | - | 83.70 | - | **98.95** | - | 98.40 |
| | CIFAR-100 | **98.10** | - | - | - | - | - | 80.76 | - | - |
| | ImageNet | **97.00** | - | - | - | - | - | - | - | - |

-

Xeon E5 CPU, and triple Nvidia GTX980 GPUs. Some time-intensive tasks were offloaded to a high-performance cluster containing multiple GPU nodes.

CIFAR-10 was used as the primary benchmark to evaluate the method. The ResNet18 (He et al., 2015) network was used and trained to achieve an accuracy of $91.39\%$ on unmodified raw images of the test set, which is a set of $10,000$ images, 1000 images from each class. JPEG2000 compression with a quality factor of $80\%$ was used to duplicate the train and test sets, and the network was re-trained on the JPEG-compressed images. When evaluated on the compressed test set, an accuracy of $90.27\%$ was obtained. The same test set was then run through the ART adversarial attack generation tool to create an FGSM attack with a perturbation strength ($\epsilon$) of 0.02, which was run through raw and JPEG networks, which resulted in accuracies of $62.72\%$ and $81.59\%$ respectively, indicating adversarial suppression. A complete summary of the effect of JPEG compression on FGSM attack across multiple datasets can be seen in Table 1.

The pre-deployment information was calculated using $10,000$ images randomly selected from the train set, and algorithm 1 was used to extract 10 class signatures from them. The $I$ parameter was set to 50 through experimentation. The evaluation was then performed on the $10,000$ image test set. These images were taken from the test set of each data set and were run as clean samples through the method. The threshold of 5 was selected such that $100\%$ of the clean images were marked as clean by the technique, with a margin added to it for the CIFAR-10 dataset. Then, each image was used to create one adversarial attack of each type, resulting in $10,000$ test images per attack. The scores given in the result summary are the percentages of images in that set marked adversarial by the method, i.e., had a $P_A$ higher than the threshold $T$.

---

[1] The method names in boldface are attack agnostic

[2] The values in boldface are the best overall result for the given dataset-attack combination

[3] The dash (-) indicates that this result has not been reported

[4] The values in italics are the best attack agnostic method result for the given dataset-attack combination, but there is a better attack-specific result

The Experiment was repeated for CIFAR-100 and a subset of TinyImageNet that consists of only 50 classes to facilitate the attack generation, which is very time/memory intensive for complex attacks such as JSMA and Elastic Net. Separate thresholds were calculated for each dataset (8.3 for CIFAR-100 and 8.7 for ImageNet). Compared with other methods, the complete result set can be seen in Table 2.

## 5 CONCLUSION

To conclude this work, we have presented a method that can detect adversarial attacks without the requirement for attack-specific training and maintain a high detection accuracy across multiple attack models without compromising the false positive rate. The results show that our method is reliable across multiple datasets and attack models, maintaining almost perfect accuracy. These characteristics make our method ideal for sensitive applications needing reliable defense against potential adversarial threats.

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
