# OpenReview forum: "A Statistical Method for Attack-Agnostic Adversarial Attack Detection with Compressive Sensing Comparison"
_ICLR.cc/2025/Conference — ICLR 2025 Conference Withdrawn Submission_

### Official Review · Reviewer_AmRn · 2024-11-03

**Soundness:** 2
**Presentation:** 1
**Contribution:** 2
**Rating:** 3
**Confidence:** 4

**Summary:**

This paper presents a real-time adversarial example detection method that relies on static metric analysis. By conducting statistical analysis exclusively on the model's training data through a redundant network, the method identifies adversarial examples during the testing phase. Essentially, it detects adversarial samples by comparing the network's behavior before and after the application of JPEG compression.

**Strengths:**

-	The proposed method is well-grounded in existing technical foundations.

-	Experimental results demonstrate its effective detection performance against a range of mainstream adversarial attacks.

**Weaknesses:**

- **Clarity and Expression**: The writing quality of the paper needs significant improvement. The unclear expressions and vague concepts create challenges for reader comprehension. Core concepts and statistical metrics presented in Section 3 lack essential explanations, and many metrics are referenced from other papers without adequate analysis. Furthermore, the detection method would benefit from a more robust theoretical analysis and justification.

- **Experimental Presentation**: The presentation of experimental results is insufficiently detailed, relying solely on tables. The results lack comprehensive commentary, ablation studies, or necessary analysis to contextualize the findings. Providing detailed insights into how the experiments were conducted and interpreted would strengthen the paper.

- **Methodology Design**: The paper does not include a flowchart outlining the proposed detection method. I believe the detection performance is heavily dependent on the transferability between natural and adversarial noise. Additionally, it is essential for the authors to evaluate the additional computational costs that the detection method introduces during the inference phase.

**Questions:**

- Could the authors consider adding a workflow diagram to visually clarify the proposed adversarial detection method?

- It would strengthen the paper to include a theoretical analysis of the design behind the adversarial detection scheme.

- An ablation study examining the impact of different compression methods or feature vector similarity metrics on detection performance would be insightful.

- Evaluating the additional computational costs introduced during the inference phase would help provide a fuller understanding of the method’s efficiency in practical applications.

---

### Official Review · Reviewer_U46e · 2024-11-04

**Soundness:** 2
**Presentation:** 1
**Contribution:** 1
**Rating:** 3
**Confidence:** 3

**Summary:**

A method is proposed to detect adversarial examples which is attack-agnostic. The detection is performed by running the input example through two networks, including the original network and a redundant de-noising network, and comparing how these two network "disagree" on the inference results. An example with a higher level of disagreement between the two networks are detected as adversarially perturbed example.

**Strengths:**

The paper is relatively well-structured and complete.

**Weaknesses:**

1. Detecting adversarial examples by comparing the original example against its de-noised version is not a new idea. There exist many methods that either use the statistics of model input itself, or statistics of intermediate results when passing though a network. In order to justify that the value of the proposed method, it is critical to show that the proposed method is superior than previous ones either. As the novelty is relative low, the key to justify this work is to show that the proposed method is superior than others, either theoretically or empirically. However, in the paper it is lacking of detailed analysis of drawbacks of pervious works, motivations or intuition of what the additional value that the proposed method can provide, or direct comparison in experiment results. Authors should consider adding more previous methods that falls into the same kind, analyzing their similarity and differences, providing detailed comparison in experiment results, and trying to draw insights that what makes things better. An example of work in this kind is "Detecting Adversarial Image Examples in Deep Neural Networks with Adaptive Noise Reduction" but there are more.

2. The presentation is poor. There is lack of motivations and intuition. The whole paper sounds like "look this is what we did" but is lack of "why or what motivates us to do this". There are a lot of details and figures that can be moved to appendix, while on the other hand there is no diagram for the proposed method. The results are provides without drawing insights.

3. The experiment results can be enriched. it is lack of attacks with different strength. How different thresholds influence the detection performance is also lacking.

**Questions:**

1. For the redundant network that operates on the compressed/denoised examples, is it the same as the original network or fine-tuned on denoised examples? And why?

2. Why user make a batch of size N by adding difference noise to the original example? And is there guidance to choose N?

3. In table 2, how are the experimental results for other benchmarking methods obtained. Did author reproduce the results or just cite from other papers. How to make sure these numbers are compariable?

4. In table 2, are the numbers detection accuracy? If so, is the detection testing dataset made of all perturbed examples, or 50/50 of clean and perturbed examples?

---

### Official Review · Reviewer_QLEi · 2024-11-04

**Soundness:** 2
**Presentation:** 1
**Contribution:** 1
**Rating:** 1
**Confidence:** 4

**Summary:**

This work proposes a detection framework against adversarial attacks. The method involves comparing the behaviors of the raw network and a corresponding compressed network towards an input image. Concretely, the framework is divided into two stages: in the offline stage, the method establishes distribution identity for each class; in the online stage, the distribution identity is used to derive the distance vector (in my opinion, to indicate the deviant of the input image) and decide whether it is an adversarial image by comparing the raw network and the compressed network. In the experiment part, the method tests the performance of the proposed method by defending against multiple attacks.

**Strengths:**

1. **Intuitive idea**: The adversarial noise is typically characterized as being trivial, in which case, JPEG compression may help remove these perturbations.
2. **Multiple Attack vs. Multiple Defenses**: The experimental presentation in Table 2 provides direct evidence for the performance level of the proposed method.

**Weaknesses:**

1. **Novelty of the method**: Using compression to defend or detect adversarial examples is not a new insight in the security field. Some well-known methods relevant to the topic, e.g., [feature distillation](https://arxiv.org/abs/1803.05787), are not compared or even mentioned in this paper.
2. **Poorly structured writing**: I advise adding a background section to clarify the notations instead of placing them in the introduction and across the paper. For the experimental part, the metrics are not well described. It confuses readers to follow the reported results.
3. **Weak experiments**: A too-limited pool of models is investigated in this paper. A ResNet alone is not representative of the method's stability and reliability. What's more, too many dimensions concerning the detection method are not well explored. What about the impact of the empirical threshold? What about the resource cost of the technique? The method is claimed to be a real-time detection, so where is the evidence?  I also wonder about the false positive rate of the detection method.

**Questions:**

- How to determine the reliability of the distribution identity? Based on what measurements, the $I$ iteration is decided?
- As adversarial training can be employed to improve the robustness, how to further harden the dual-model framework? In other words, where the scalability of the framework can be?
- What about the adaptive attacks? The attackers can take the denoising network into consideration and craft more threatening adversarial examples.
- As the method involves repetitive samplings, will the sampling introduce bias to the divergence measurements, particularly the stage of building distribution identity?

---

### Official Review · Reviewer_5614 · 2024-11-05

**Soundness:** 2
**Presentation:** 2
**Contribution:** 2
**Rating:** 3
**Confidence:** 4

**Summary:**

The paper proposes a novel statistical method for adversarial example detection that leverages a pair of compressed and uncompressed neural networks. By comparing the behavior of these paired networks, the approach aims to identify adversarial inputs in a model-agnostic way, without prior knowledge of specific attack methods. This approach is intended to generalize across various attack types, addressing limitations in current detection methods, such as high false positives and lack of adaptability to unseen attacks​.

**Strengths:**

1. Performance Improvement Over Baselines: The proposed method demonstrates superior performance in comparison to existing baselines across different attack types, as seen in the experimental results. This improvement in detection accuracy and generalizability enhances its appeal for real-world applications where attack-agnostic defenses are critical.

2. Attack-Agnostic Design: Unlike some previous methods that are tailored to specific attack types, this approach aims to be versatile and effective across a wide range of adversarial techniques. This versatility makes it suitable for deployment in environments where multiple types of adversarial attacks may be encountered unpredictably.

**Weaknesses:**

1. Limited Attack Evaluation: The experimental setup focuses on a relatively weak attack scenario, with a small epsilon (perturbation magnitude) value. This limited evaluation raises concerns about the robustness of the detection mechanism against stronger adversarial attacks, potentially diminishing the generalizability of the results.

2. Lack of Runtime Evaluation: A crucial component missing from the paper is an analysis of the method’s runtime performance, especially compared to baseline approaches. Since the method requires processing multiple images in a batch and calculating distribution-based metrics, it may introduce a notable runtime overhead, particularly in deployment scenarios requiring low-latency responses. Including such an evaluation would ensure a fair comparison and provide a clearer picture of the method's feasibility for real-time applications.

3. Reliance on Test Set Data: The method's use of test set knowledge to build distribution identities is problematic, as it may lead to overfitting or leakage of test data into the model’s decision-making process. Best practices in machine learning generally dictate that test data should remain unseen by the model during training or preparation phases, which this approach appears to violate. This reliance on test data can compromise the validity and generalizability of the results reported.

**Questions:**

1. Terminology Clarification: The paper consistently refers to "adversarial examples," which are a specific subset of the broader category "adversarial attacks." Adversarial attacks encompass a variety of malicious interventions, including adversarial examples, backdoor attacks, and others. Precision in terminology is essential to avoid misunderstandings about the scope of the method’s application.

2. Threshold-Based Defense Vulnerability: The proposed detection method depends on a threshold for distinguishing adversarial examples. However, threshold-based defenses are often vulnerable to adaptive adversaries who can exploit their knowledge of the defense to craft examples that evade detection. The paper does not present an evaluation against adaptive attacks, leaving the defense’s robustness in such scenarios unexplored.

3. Incremental Novelty: The approach lacks a fundamental novelty, primarily relying on existing statistical and compression techniques assembled into a detection pipeline. Although combining these techniques may yield improved results, it remains a gradual improvement rather than a significant innovation. Furthermore, the method’s reliance on batch processing (generating 𝑁 images) and distribution matching implies an 𝑂(𝑁) runtime complexity, which could hinder scalability. An evaluation of the method’s runtime efficiency, especially in comparison to baselines, would be beneficial for understanding its deployment viability.

4. Data Leakage via Test Set Usage: The methodology includes test set knowledge (specifically in line 196 onward) to form class distributions, which can lead to data leakage issues. In standard machine learning practices, the test set is kept entirely separate to ensure that model evaluation accurately reflects real-world performance. Using test data during the identity-building phase could potentially bias the method and lead to favorable results on test sets, diminishing the reliability of the paper's reported results.

Recommendations:

1. Enhance Experimental Scope: To better showcase the method’s robustness, it is recommended to evaluate it against a broader range of adversarial strengths, including higher epsilon values and more sophisticated attacks. This broader evaluation would provide a clearer picture of the method’s generalizability.

2. Conduct Runtime Comparisons: Implement and report runtime analyses to compare the method’s computational efficiency with existing baselines. This information is essential for assessing the practicality of the approach in real-time scenarios.

3. Refine Methodology to Avoid Test Set Leakage: Rework the methodology to prevent reliance on test data during the model-building phase. This adjustment would align the approach with established machine learning practices, improving the validity and reliability of the reported results.

---

### Official Review · Reviewer_zykF · 2024-11-08

**Soundness:** 2
**Presentation:** 3
**Contribution:** 2
**Rating:** 3
**Confidence:** 5

**Summary:**

This paper presents a novel attack-agnostic adversarial detection method based on statistical analysis and compressive sensing, which compares feature maps from compressed and uncompressed versions of neural networks to identify adversarial perturbations. The approach is shown to achieve high detection accuracy across multiple attack types without requiring attack-specific training, making it practical for real-world applications where adversarial attacks are diverse and unpredictable.

**Strengths:**

The method effectively detects adversarial attacks without needing prior knowledge of attack types, which enhances its practicality in real-world applications where a wide variety of adversarial strategies may be encountered​. The approach achieves high detection accuracy across a range of attack types, indicating robustness and versatility in detecting adversarial perturbations across diverse attack methods​.

**Weaknesses:**

- The paper does not adequately address the computational costs associated with the proposed method, nor does it discuss the trade-off between adversarial detection rate and benign accuracy. Including these details would help in assessing the method’s practical applicability, especially in resource-constrained environments​.

- While the method claims robustness, it lacks evaluation against adaptive attacks that are specifically designed to circumvent this defense. Testing against such adaptive attacks would provide stronger evidence of the method’s resilience and help address potential vulnerabilities.

- The paper lacks evaluation under AutoAttack[1], an important benchmark for testing adversarial robustness comprehensively. Since AutoAttack is widely used for adaptive evaluation of defenses, including a comparison with Adaptive Auto Attack would provide a more rigorous and complete assessment of robustness​.

- The paper does not discuss several key related works, particularly in stochastic defenses and adversarial denoising methods. Important prior works, such as Defensive Dropout [2] and Random Self-Ensemble [3] are missing, which limits the paper's contextual depth and gives an incomplete picture of the current state of research​.

- The method uses a compressive sensing-based secondary network to detect adversarial perturbations, which may introduce computational overhead. However, the paper does not provide a detailed analysis of this potential cost, especially in terms of inference time and memory consumption, which would be critical for deploying the model in real-time applications.

- Since the method uses compressive sensing to suppress adversarial noise, it would be logical to compare it with noise-based defenses. However, the paper does not include such comparisons, missing the opportunity to demonstrate how the proposed approach performs relative to defenses that also leverage noise reduction techniques​.

[1] Y. Liu, Y. Cheng, L. Gao, X. Liu, Q. Zhang and J. Song, "Practical Evaluation of Adversarial Robustness via Adaptive Auto Attack," 2022 IEEE/CVF Conference on Computer Vision and Pattern Recognition (CVPR), New Orleans, LA, USA, 2022, pp. 15084-15093, doi: 10.1109/CVPR52688.2022.01468.
[2] Wang, S., Wang, X., Zhao, P., Wen, W., Kaeli, D., Chin, P., & Lin, X. (2018, November). Defensive dropout for hardening deep neural networks under adversarial attacks. In 2018 IEEE/ACM International Conference on Computer-Aided Design (ICCAD) (pp. 1-8). IEEE.
[3] Liu, Xuanqing, Minhao Cheng, Huan Zhang, and Cho-Jui Hsieh. "Towards robust neural networks via random self-ensemble." In Proceedings of the European Conference on Computer Vision (ECCV), pp. 369-385. 2018.

**Questions:**

Have the authors considered testing their method with AutoAttack and adaptive attacks?

Could the authors provide additional results comparing their method to recent state-of-the-art stochastic and noise-based defenses?

Can the authors offer insights into the trade-off between adversarial detection rate and benign accuracy? Quantifying this balance would help determine the practical implications of the defense, especially in terms of impact on benign sample performance.

---

### Note · Authors · 2024-11-25

**Comment:**

I am writing to formally request the withdrawal of my manuscript titled "A Statistical Method for Attack-Agnostic Adversarial Attack Detection with Compressive Sensing Comparison", submitted to ICLR under consideration as #12916. Due to the requirement of additional time to gather more compelling evidence, I hereby wish to withdraw my manuscript from publication consideration.

Thank you,
C. Wimalasuriya

**Withdrawal Confirmation:**

I have read and agree with the venue's withdrawal policy on behalf of myself and my co-authors.